# Edible Flower Species as a Promising Source of Specialized Metabolites

**DOI:** 10.3390/plants11192529

**Published:** 2022-09-27

**Authors:** Mia Dujmović, Sanja Radman, Nevena Opačić, Sanja Fabek Uher, Vida Mikuličin, Sandra Voća, Jana Šic Žlabur

**Affiliations:** 1Department of Agricultural Technology, Storage and Transport, University of Zagreb Faculty of Agriculture, Svetošimunska cesta 25, 10000 Zagreb, Croatia; 2Department of Vegetable Crops, University of Zagreb Faculty of Agriculture, Svetošimunska cesta 25, 10000 Zagreb, Croatia

**Keywords:** polyphenolics, ascorbic acid, chlorophylls, carotenoids, antioxidant capacity, new food

## Abstract

Eating habits are changing over time and new innovative nutrient-rich foods will play a great role in the future. Awareness of the importance of a healthy diet is growing, so consumers are looking for new creative food products rich in phytochemicals, i.e., specialized metabolites (SM). The consumption of fruits, vegetables and aromatic species occupies an important place in the daily diet, but different edible flower species are still neglected and unexplored. Flowers are rich in SM, have strong antioxidant capacities and also possess significant functional and biological values with favorable impacts on human health. The main aim of this study was to evaluate the content of SM and the antioxidant capacities of the edible flower species: *Calendula officinalis* L. (common marigold), *Tagetes erecta* L. (African marigold), *Tropaeolum majus* L. (nasturtium), *Cucurbita pepo* L. convar. *giromontiina* (zucchini) and *Centaurea cyanus* L. (cornflower). The obtained results showed the highest content of ascorbic acid (129.70 mg/100 g fw) and anthocyanins (1012.09 mg/kg) recorded for cornflower, phenolic compounds (898.19 mg GAE/100 g fw) and carotenoids (0.58 mg/g) for African marigold and total chlorophylls (0.75 mg/g) for common marigold. In addition to the esthetic impression of the food, they represent an important source of SM and thus can have a significant impact if incorporated in the daily diet.

## 1. Introduction

Nowadays, the demand in the food market is becoming more and more diverse. The food industry is constantly developing and creating new creative food products to attract consumers who are increasingly interested in trying new foods and accessing new, nutritious and healthy food sources. In addition, the awareness of the importance of a proper and sustainable diet rich in phytochemicals, i.e., specialized metabolites (SM), and other antioxidants that can positively affect human health is growing. Thus, some of the new food sources that are becoming more popularized among both nutritionists and food manufacturers include different edible flower species.

Specialized plant metabolites are not vital for maintaining basic physiological and life processes, but perform specific functions such as responses to abiotic and biotic stresses, protection from pathogens or UV radiation and contribution to taste, smell, color, etc. Plants produce a huge amount of specialized metabolites with various specific roles, thanks to which they have a number of applications for human use and many industrial processes [1].

The fruits and leaves of a number of fruits and vegetables species are common ingredients in healthy daily meals, while at the same time the use of edible flower species for food purposes is strongly neglected and still relatively unknown to consumers. A flower, blossom or bloom is the reproductive structure of flowering plants (division Angiospermae), whose biological function is to facilitate plant reproduction and produce fruits containing one or more seeds [2]. However, regardless of the unquestionable role in the reproductive life of the plant, flowers are also a rich source of various SM such as phenols, ascorbic acid (AsA) and pigments, which highlight this type of food as a nutritious, valuable and highly appreciated antioxidant source [3,4]. The chemical composition of edible flowers is heterogeneous, but the most abundant chemical compounds are phenols, of which the most common and the most important are phenolic acids, flavonoids and anthocyanins, followed by carotenoids (carotenes, xantophylls) and chlorophylls; alkaloids, betalains, betacyanins and betaxanthins; nitrogen-containing and organosulfur compounds, etc., [5]. Thanks to the rich content of various SM, edible flower species can have significant biological and functional values, such as: antioxidant [6], immunomodulatory [7], anticancer [8,9], antiviral, antibacterial, anti-inflammatory, antiallergic, cardioprotective, hepatoprotective, neuroprotective and antimalarial properties [10,11]. All the above mentioned, coupled with the specific chemical and nutritional contents of edible flower species, highlight this food source as very important for humans providing numerous benefits in the prevention and treatment of many chronic diseases [12,13].

Flowering plants have had a significant role throughout human history [14] and have been present in everyday life ever since. They are used for indoor and outdoor decorations, inspiration in art, mood enhancement, air purification, elimination of bad odors, tradition, herbal medicine [15], honey production [16] and can be a source of essential oils [17], natural dyes and food colors [18]. Thanks to their esthetic value and recognized substances with health-promoting properties, some flower species have become a part of the human diet. Benvenuti and Mazzoncini [19] have listed 88 flower species rich in SM that have been studied as nutraceutical foods in the last two decades. Edible flowers add a fresh, exotic and delicate flavor and visual appeal to dishes; they are increasingly used in gourmet cuisine [20]. Flowers are part of many regional cuisines, including Asian, European and Middle Eastern [21]. So far, theyhave been mainly used as food decorations in desserts, cakes, beverages and infusion blends, as candied flowers, in the preparation of raw salads, fried dishes and other meals and for flavoring butters, oils and syrups.

Generally, regardless of their final use, the flowers are harvested when the blossoms are open. For example, perfume and cosmetics industries largely extract many volatile aromatic molecules and pigments synthesized by flowering plants, mostly accumulated during the flowering period [22]. The cut flower industry produces fresh open flowers for floral arrangements and decorations. Although most edible species are also collected and consumed during flowering, some are used at the budding stage, depending on the species. While dandelion’s (*Taraxacum officinale* Web.) sweet and honey-flavored flower buds can be eaten raw or cooked [23], young flower buds of the caper bush (*Capparis spinosa* L.), known as capers, are mostly pickled and used as a spice and addition to sauces, salads or pizzas [24]. Edible flower species can be obtained from cultivation or collected in nature (wild populations); while soil, water and air pollution are the main factors to consider when collecting flowers from the wild. Usually, flower species are cultivated for decorative purposes with the use of pesticides, but if the ultimate goal is use for nutritional purposes, then it is very important to know their origin and the agricultural measures used in cultivation. From an agronomic and health point of view, organic cultivation is the only way to control the traceability, safety and quality of edible flowers, and this type of produced flowers is the most suitable for consumption [25]. Organic cultivation systems are recommended for the growing of edible flowers [26] because these species mainly have a natural resistance to pests and therefore do not require significant use of pest control products or pesticides, which also makes them more suitable for consumers since the flowers are consumed fresh. In addition, the cultivation of flowering plants can have a positive impact on agricultural crops, which is important for the preservation of biodiversity and has numerous advantages for organic farming in general. Some ornamental plant species, such as: African marigold (*Tagetes erecta* L.), nasturtium (*Tropaeolum majus* L.), pyrethrum daisy (*Tanacetum cinerariaefolium* L.), tansy (*Tanacetum vulgare* L.) and chamomile (*Matricaria chamomilla* L.) are great companions for other plants; they can significantly increase soil fertility and act as weed suppressants, repellents (natural insecticides, so there is no need for use of chemical insecticides) or attractants for beneficial insects (bees) [27,28]. Although there is a wide variety of flower species, not all of them are edible and some are even poisonous [14], such as azaleas (*Rhododendron* spp.) [29], daffodils (*Narcissus* spp.) [30], foxgloves (*Digitalis* spp.) [31] and hydrangeas (*Hydrangea* spp.) [32], therefore it is also very important to know which species are suitable for human consumption.

Edible flower species are also commonly grown in private gardens, on terraces or balconies, so they can be an easily accessible source of phytochemicals even in smaller areas and limited spaces, such as in urban areas [33]. Therefore, it is important to investigate the nutritional value, SM content and antioxidant capacity of edible flower species. The aim of this study was to determine the SM content and antioxidant capacity of the following organically cultivated edible flower species: *Calendula officinalis* L. (common marigold), *Tagetes erecta* L. (African marigold), *Tropaeolum majus* L. (nasturtium), *Cucurbita pepo* L. convar. *giromontiina* (zucchini) and *Centaurea cyanus* L. (cornflower). All of these selected species are ornamental herbaceous annuals that are widely distributed but have different applications. Previous research has confirmed that these species are edible [34], but their extensive biological and culinary potential has yet to be discovered; thus, the results of this study will provide new data on the nutritional and antioxidant capacity values of these flowers. The novelty of this research compared with others is that edible flowers from organic cultivation were investigated, thus promoting organic production but also raising awareness of the importance of the origin of the plant material.

## 2. Results

In the result section the chromaticity parameters (Table 1), total dry matter content (Figure 1), SM content (Table 2), pigment compounds (Table 3) and antioxidant capacity (Figure 2) of analyzed fresh edible flower species are presented.

### 2.1. Chromaticity Parameters and Total Dry Matter Content of Fresh Edible Flowers

The chromaticity parameters of analyzed fresh edible flowers are listed in Table 1. The L* values ranged from 36.12 to 70.90, depending on the flower species. Considering a* values, a red color is present in all species, with the highest content observed for nasturtium, common marigold and African marigold. A negative b* value was measured for cornflower, while the remaining flowers had positive b* values, indicating the presence of a yellow color.

Figure 1 shows the total dry matter content (DM %) of the analyzed edible flower species. According to the conducted statistical analysis, all flower samples differ significantly in DM content. The obtained results show that the highest DM was determined for the flower samples of the species cornflower (34.37%), followed by African marigold (19.64%), common marigold (17.94%) and nasturtium (12.43%). The lowest DM was determined for zucchini flowers (10.45%).

### 2.2. Specialized Metabolites of Fresh Edible Flowers

The results of SM of the edible flower species are shown in Table 2. The highest ascorbic acid (AsA) content (129.7 mg/100 g fw) was determined for the cornflower, while the lowest (ranging from 25.46 to 36.69 mg/100 g fw) for common marigold, zucchini and African marigold. The high AsA content was also recorded for the nasturtium flower species (77.56 mg/100 g fw).

According to the statistical analysis, all samples differed significantly in their content of polyphenolic compounds (Table 2). The highest content of total phenolics (TPC), total flavonoids (TFC) and total non-flavonoids (TNFC) was observed for African marigold (898.19 mg GAE/100 g fw TPC; 391.09 mg CTH/100 g fw TFC; 507.11 mg GAE/100 g fw TNFC), followed by cornflower (647.09 mg GAE/100 g fw TPC; 264.90 mg CTH/100 g fw TFC; 382.20 mg GAE/100 g fw TNFC). Common marigold had higher TPC and TNFC (379.38 mg GAE/100 g fw TPC; 213.89 mg GAE/100 g fw TNFC) but lower TFC than nasturtium. The lowest content of all polyphenolic compounds (including TPC, TFC and TNFC) was determined for zucchini flowers.

The results of the analyzed pigment compounds: chlorophyll a (Chl_a), chlorophyll b (Chl_b), total chlorophyll (TCh), total carotenoid (TCa) and total anthocyanin (TAC) content are presented in Table 3. The highest Chl_a (0.32 mg/g), Chl_b (0.43 mg/g) and TCh content (0.75 mg/g) were determined for common marigold flowers, followed by African marigold with 0.12 mg/g of Chl_a, 0.20 mg/g of Chl_b and 0.32 mg/g of TCh. Very low levels of Chl_a, Chl_b and TCh were detected in nasturtium and zucchini flowers compared with the previously mentioned species. As for TCa, the highest value was noticed for African marigold (0.58 mg/g), followed by common marigold flowers (0.42 mg/g), while nasturtium and zucchini had the same and significantly lower values of TCa (0.28 mg/g). Cornflower is the only flower species for which anthocyanins were recorded; this value was 1012.09 mg/kg.

### 2.3. Antioxidant Capacity of Fresh Edible Flowers

The results of the antioxidant capacity of the studied edible flower species are shown in Figure 2. The flower species common marigold (2489.04 µmol TE/L), African marigold (2500.22 µmol TE/L), nasturtium (2502.85 µmol TE/L) and cornflower (2497.53 µmol TE/L) show the highest values, which differ significantly from those of zucchini with the lowest antioxidant capacity value of 2256.29 µmol TE/L.

## 3. Discussion

Almost all flowering plants have evolved colorful and noticeable corollas that stand out from the green biomass as a method to attract pollinators (mainly bees) [19]. The color of flowers is particularly attributed to the presence of carotenoids, flavonoids and anthocyanins [35], so the coloration of individual plant species is the first indicator of pigment composition and is of great importance for their physical properties. Color is one of the most important visual factors associated by consumers with the attractiveness and quality of the final food product. And some preservation technologies, such as the use of low temperature [36], control of ethylene synthesis, microbial proliferation and respiration and modified atmosphere packaging are used to prolongate the shelf-life of edible flowers [37]. Chromaticity parameters analyzed within this study show the highest L* value measured for common marigold, indicating the brightest flowers, while for African marigold and cornflower the lowest L* values were measured, indicating darker flowers. Cornflower was expected to have the lowest L* value since its blue–purple flowers are darker compared with other flower species. The highest a* values were observed for common marigold, African marigold and nasturtium, indicating a higher presence of red color considering cornflower and zucchini flowers. As expected, a negative b* value was measured for the cornflower, indicating the presence of blue color, and the highest positive b* value was measured for the common marigold, indicating the strongest yellow coloration. Socha et al. [38] also measured chromaticity parameters for the same flower species: common marigold (L* 68.56, a* 14.22, b* 42.44), African marigold (L* 54.44, a* 22.49, b* 28.45), nasturtium (L* 54.40, a* 14.59, b* 23.48), zucchini (L* 68.96, a* 10.58, b* 34.39) and cornflower (L* 45.20, a* 2.51, b* 15.62). It can be concluded that the highest b* values were measured for common marigold and negative values for cornflower in both research.

Total dry matter content (DM) is an indicator of the nutritional food quality, i.e., available nutrients such as carbohydrates, fats, proteins, fibers, vitamins, minerals and SM. In general, foods with a higher percentage of dry matter content have a higher nutritional value [38,39]. Both Mlcek et al. [40] and Demasi et al. [41] detected a higher DM content in common marigold (8.98% DM and 13.7% DM) than nasturtium (7.38% DM and 8.9% DM), confirming the results of this study. Demasi et al. [41] also recorded a high DM content in cornflower (26.4% DM); which is a slightly lower value compared with the results of this study. Within the edible flower species analyzed in this study, the lowest content of DM, i.e., the highest water content, was determined for zucchini flowers, confirming the fact of difficult storage and rapid decay after the harvest of the flowers of this species. Possible reasons for the wide range of DM content among different edible flower species and the other literature data may be due to various environmental factors (temperature, precipitation, pollution, etc.) [42], agronomic measures (fertilization, sowing density, etc.) [43] or genetic characteristics [44]. In this study, species from the *Asteraceae* family (cornflower, African marigold and common marigold) generally had higher DM accumulation than species from other families (*Cucurbitaceae* and *Tropaeolaceae*). In general, species with lower DM, i.e., higher water content (as in the case of nasturtium and zucchini), have lower storability, making them more susceptible to deterioration and to a reduction in nutritional quality. Additionally, during handling in the food supply chain, given the DM content, it is preferable to pack such plant material in specific packaging materials (biodegradable materials, perforated plastics, transparent polyethylene trays) in a controlled atmosphere (optimal CO_2_ and O_2_ ratio) that will ensure adequate transpiration and respiration of the product, and thus extend product quality over a longer period of time [45].

In addition to its important antioxidant function, ascorbic acid (AsA) is responsible for a number of other biochemical functions in the human body, such as the prevention of anemia, absorption of inorganically bound iron [46], biosynthesis of collagen, regulation of many genes responsible for tumor growth, energy metabolism, apoptosis, control of infections and inflammations and treatment of neurodegenerative diseases [47]. Humans cannot synthesize AsA by themselves [48], so to achieve a regular and adequate intake, a daily consumption of fresh fruits, vegetables and other plant sources rich in AsA is recommended [49]. According to the results of this research, some edible flower species can also represent a good source of AsA. For example, cornflower stands out with the highest AsA content, even 67% higher than that of nasturtium. The remaining three tested species had the lowest AsA content, ranging from 25.46 to 36.70 mg/100 g fw. Considering the obtained results, all analyzed edible flower species are valuable sources of AsA, which can be associated with the other literature data that also highlight some edible flower species (*Ageratum houstonianum* Mill, *Tagetes lemmonii* A. Gray, *Salvia dorisiana* Standl, *Pelargonium odoratissimum* L.) with high AsA contents [36], thus emphasizing these species as unfairly neglected in terms of AsA. Other studies recorded AsA contents of 71.5 mg/100 g fw for nasturtium [50] and 57.3–91.0 mg/100 g for African marigold [51]. When comparing the edible flower species from this study with other food sources popular in AsA content, such as yellow bell pepper (*Capsicum annuum* L.) (159.61 mg/100 g fw) [52], kiwifruit (*Actinidia deliciosa* (A.Chev.) C.F.Liang and A.R.Ferguson) (65.5 mg/100 g fw) [53], papaya (*Carica papaya* L.) (61.8 mg/100 g fw) [54], pomegranate (*Punica granatum* L.) (60.78 mg/100 g) [55], strawberries (*Fragaria* spp.) (58.8 mg/100 g fw), oranges (*Citrus* × *sinensis* L.) (53.2–58.3 mg/100 g fw) [56,57], lemon (*Citrus limon* (L.) Osbeck) (43.96 mg/100 g) [57] or tomatoes (*Solanum lycopersicum* L.) (22.4 to 28.4 mg/100 g fw) [58], etc., it can be concluded that these species can be implemented in the daily diet as very valuable sources of AsA. Moreover, the presented results of this study show that cornflower has an extremely high content of AsA compared with other good AsA sources, which makes it a potentially precious source for the food, pharmaceutical or cosmetic industries. It is important to note that AsA biosynthesis in plants is strongly influenced by different factors such as photosynthesis rate, genetic predisposition [59], stress exposure [60,61], phenophase, etc. AsA is also involved in a number of important metabolic functions in the plant organism, including detoxification of reactive oxygen species (ROS), and is a cofactor for the biosynthesis of some plant hormones (ethylene, gibberellic acid and abscisic acid), which affect the regulation of developmental processes, including senescence and flower formation [62].

Phenolic compounds are considered to be one of the strongest antioxidants with the potential to reduce cancer risk, decrease chronic diseases and provide other health benefits [63,64]. The highest TPC, TFC and TNFC content was observed for African marigold, while the lowest content for zucchini flowers. Socha et al. [34] investigated edible flowers grown in Polland, including the species analyzed in this study (common marigold, African marigold, nasturtium, zucchini and cornflower). The results also showed the highest TPC for African marigold, having a 420% higher content than common marigold. While the highest TF content among the mentioned species was again detected for African marigold, the value was 95% lower for nasturtium. Mlcek et al. [40] detected the same TPC for nasturtium and common marigold, while a higher TFC was observed for common marigold. Different data were obtained by Demasi et al. [41], where cornflower and nasturtium did not statistically differ and had 100% and 88% higher TPC than common marigold. In a study conducted by Lockowandt et al. [65], 4 phenolic acid derivatives, 12 non-anthocyanin flavonoids and 4 anthocyanins were identified in cornflower. Plant phenols play various roles in the plant organism, such as growth control, mechanical support, attracting, signaling and protective functions [66], flavor formation [67] and defense against biotic and abiotic factors [68]. Some studies indicate the significant increase in TPC from the bud to the senescent stage [69]. The increase of total phenols at the flowering phase may be due to the failure of redistribution of the phenols to the developing parts or by the increased synthesis of phenols toward senescence as part of the defense mechanism [62], thus mature flowers are possibly a desirable source of these compounds.

Plant pigments, including chlorophylls, carotenoids and anthocyanins, determine the attractive color, play a variety of roles throughout the whole plant life cycle and are characterized by powerful antioxidant [70] and anti-inflammatory activity [11,71]. There is research confirming that chlorophyll derivatives have multiple potential health benefits [8], may be protective agents against chronic diseases and have cytostatic and cytotoxic activities against tumor cells [72], therefore it is recommended to include them in the daily diet through pigment-rich foods. At the early developmental stages, the petals of many flowering plants contain chlorophylls and, as they mature, the chlorophyll content decreases and other pigments accumulate [73]. Since a small amount of chlorophyll remains in flowers during maturation and SM can have biological effects even when present in small quantities thanks to mutual synergistic interactions [74], it was important to examine the presence of chlorophylls in flower samples. In this study, the highest TCh value was recorded for common marigold (for example, African marigold had a 57% lower TCh value compared with the common marigold), while the lowest content was observed for nasturtium, zucchini and cornflower. Surprisingly, the results showed that the TCh content was 79% higher than the TCa content for common marigold, indicating that some colorful flowers may be a better source of chlorophyll than other pigments. As for carotenoids, the highest value was determined for African marigold flowers, followed by common marigold, while nasturtium and zucchini had the lowest recorded TCa values. Several studies have shown that, among edible flower species, carotenoid rich sources may also be French marigold (*Tagetes patula* L.), African marigold [75] and nasturtium [76]. Cornflower was shown to be rich in anthocyanins, as suggested by the intensive blue color of the flowers and as confirmed by the obtained results, which is in accordance with the data of Fernandes et al. [77], who also noted that the cornflower’s fresh petals are rich in anthocyanins. Anthocyanins are one of the most important compounds from the flavonoid group, since they are a strong antioxidants; thus, species rich in anthocyanins generally have potential health value [78].

Antioxidants are the major scavengers of reactive oxygen (ROS) and nitrogen species (RNS) [6,79], with a crucial role in preventing oxidation processes in cells, which are mostly caused by harmful free radicals that accumulate in cells due to exposure to stress, negative environmental influences, toxins, etc. [80]. These compounds are of great importance for human health because they can prevent many chronic diseases such as tumors, diabetes, neurodegenerative, cardiovascular and other diseases [80]. The antioxidant capacity of flowers derives from their richness in SM and is proportional to the content of carotenoids, flavonoids (especially anthocyanins), phenolic acids, vitamins and essential oils [81,82]. All mentioned SM investigated in this study (AsA, phenolics and pigments) are well recognized antioxidants. Fruits, vegetables, cereals and medicinal plants are known for a while to contain a number of compounds with strong antioxidant capacities [83], but some previous research stated some flower species also have strong antioxidant capacities such as *Rosa rugosa* Thunb., *Paeonia lactiflora* Pall. [84], *Nasturtium officinale* W.T. Aiton [85] and *Robinia hispida* L. [86]. A high antioxidant capacity has been found in the floral tissues not only before their ingestion but also after the digestive processes, highlighting the long-lasting bioactive effect of the different SM [87]. Demasi et al. [41] recorded a high antioxidant capacity in nasturtium and cornflower, while Takahashi et al. [14] also detected the significant potential of cornflower as a powerful antioxidant agent. More than 20 antioxidant compounds were found in zucchini flowers within the study of authors Takahashi et al. [14], while other researchers also suggested the valuable antioxidant capacity of zucchini flowers [88]. In this study, zucchini flowers showed the lowest antioxidant capacity, as expected, since they had the lowest content of all analyzed SM. Other researched flower species (common marigold, African marigold, nasturtium and cornflower) in general had an 11% higher antioxidant capacity compared with the zucchini flower sample. Cornflower had the same high antioxidant capacity as common marigold, African marigold and nasturtium, suggesting that, in addition to phenols, anthocyanins are also significant antioxidants presented in this species. Considering the high antioxidant potential of the analyzed species, they can be used in the production of antioxidant food supplements for preventing cell damage caused by free radicals and thus can take a significant place as a valuable nutritional source.

## 4. Materials and Methods

### 4.1. Plant Material

Five cultivated species of fresh edible flowers were analyzed in this study: common marigold (*Calendula officinalis* L.), African marigold (*Tagetes erecta* L.), nasturtium (*Tropaeolum majus* L.), zucchini (*Cucurbita pepo* L. convar. *giromontiina*) and cornflower (*Centaurea cyanus* L.). All flower species were grown in accordance with organic farming requirements on the family farm Horg near Pula, Croatia (Vinkuran 44°50′ N 13°52′ E). These are annual plants; the zucchini seedlings were planted and the remaining plants were sown directly in the open field in the spring of 2021. Flowers were harvested on June 22, 2021 at the full bloom stage, except for the zucchini flowers, which were harvested at the preopening flower stage (as they are intended for the market at this stage). The flowers were picked manually, randomly per plot, in the early morning hours during dry weather. A total of 300 g of each flower species was collected and immediately after harvest stored in polystyrene packaging in the refrigerator (at 4 °C and a relative humidity of 76%) for 24 h until they were transported to the laboratory for further planned chemical analyses. All analyses were performed at the Department of Agricultural Technology, Storage and Transport, Faculty of Agriculture, University of Zagreb.

### 4.2. Determination of Chromaticity Parameters and Total Dry Matter Content

The chromaticity parameters of the fresh flower species (L*, a*, b*, C, h°) were measured using a colorimeter (ColorTec PCM+, PCE Instruments, Southampton, UK) according to the CIELAB method. The L* value indicates lightness of sample, from 0 (indicating black) to 100 (indicating white). The value of the a* parameter indicates the intensity of red or green, so negative values (-a*) represent the presence of green and positive values (+a*) represent the presence of red color. The b* parameter represents the blue–yellow contrast, with negative values (-b*) toward blue and positive (+b*) toward yellow color [89].

Total dry matter content (DM, %) was determined by drying at 105 °C to constant mass using a standard laboratory protocol according to AOAC [90]. Total dry matter content was expressed as a percentage using Equation (1):(1)$$\mathrm{Dry}\;\mathrm{matter}\;\mathrm{content}\;\left(\%\right)=\frac{m_{2}-m_{0}}{m_{1}-m_{0}}\times 100$$
where m_0_ (g) is the mass of glassware; m_1_ (g) is the mass of glassware with the flower samples before drying; m_2_ (g) is the mass of glassware with the flower samples after drying.

### 4.3. Determination of Specialized Metabolites Content

The following compounds were determined from the group of SM: ascorbic acid content (AsA), total phenols (TPC), total flavonoids (TFC), total non-flavonoids (TNFC), total anthocyanins content (TAC) and pigment compounds (total chlorophylls, chlorophyll a, chlorophyll b, total carotenoids).

The ascorbic acid (AsA) content was determined by titration with 2,6-dichloroindophenol (DCPIP) according to the standard AOAC method [91]. For each flower sample, 10 g ± 0.01 of the plant material was homogenized with 100 mL of 2% (*v/v*) oxalic acid. The prepared solution was then filtered through Whatman filter paper and obtained in a volume of 10 mL. The filtrate was titrated with freshly prepared DCPIP until the appearance of pink coloration. The final AsA content was expressed as mg/100 g fresh weight (fw), and calculated according to Equation (2):(2)$$\mathrm{AsA}\;\left(\mathrm{mg}/\left(100\;g\right)\;\mathrm{fw}\right)=\frac{V\times F}{D}\times 100$$
where V is the volume (mL) of DCPIP; F is the factor of DCPIP; D is the sample mass (g) used for titration.

Total polyphenolic compounds, including total phenolics (TPC), total flavonoids (TFC) and total non-flavonoids (TNFC), were determined according to a method previously described by Ough and Amerine [92]. Final contents were determined spectrophotometrically (Shimadzu, 1900i, Kyoto, Japan) at a wavelength of 750 nm using distilled water as a blank. The method is based on the color reaction that phenols develop with the Folin–Ciocalteu (FC) reagent. For each flower species, 10 g ± 0.01 of the sample was homogenized with 40 mL of 80% EtOH (*v*/*v*), heated to boiling point and refluxed for 10 min. The samples were then filtered through Whatman paper and refluxed again for 10 min by adding another 50 mL of 80% EtOH (*v*/*v*). The samples were again filtered into a 100 mL volumetric flask and made up to the mark with 80% EtOH (*v*/*v*). Obtained extracts were used for the reaction with the FC reagent. To a 50 mL volumetric flask was added as follows: 0.5 mL of ethanolic flower extracts, 30 mL of distilled water (dH_2_O), 2.5 mL of the freshly prepared FC reagent (1:2, *v*/*v*), 7.5 mL of saturated sodium carbonate solution (Na_2_CO_3_) and made up to the mark with dH_2_O. The prepared reaction was allowed to stand for 2 h at ambient temperature, with periodically shaking, and, afterward, the absorbances of the solution were measured.

The same ethanolic extracts prepared for TPC were also used for TNFC analysis. TNFC determination was performed according to the following procedure: 10 mL of the ethanolic extract, 5 mL of HCl in EtOH (1:4, *v*/*v*) and 5 mL of formaldehyde (p.a.) were added to a 25 mL volumetric flask. The prepared samples were blown through with nitrogen (N_2_) and left in a dark place at room temperature. After 24 h, the samples were filtered and the same FC reaction as for TPC was performed. TFC was calculated as the difference between the amount of TPC and TNFC. Gallic acid and catechol were used as external standards, and TPC, TNFC and TFC were expressed as milligrams of gallic acid (GAE) and catechol (CTH) equivalents per 100 g fresh weight (fw).

Total anthocyanin content (TAC, mg/kg fw) was determined by bleaching with bisulfite using the method described by Ough and Amerine [92]. The method is based on the binding of bisulfite ions to anthocyanins, resulting in discoloration of the sample. The amount of 2 g of the flower sample was weighed into the cuvette and 2 mL of 0.1% HCl (96% EtOH, *v*/*v*) and 40 mL of 2% HCl (*v*/*v*) were added. After 10 min of centrifugation at 4500 rpm, the clear portion was separated and 10 mL of the sample was transferred into test tubes. To one test tube, 4 mL of distilled water, and to the other, 4 mL of 15% NaHSO_3_ (*v*/*v*), were added. After incubation of 15 min at ambient temperature, the differences in absorbance were measured spectrophotometrically at 520 nm (Shimadzu, 1900i, Kyoto, Japan). For the blank probe, 2% HCl (*v*/*v*) was used. The anthocyanin content was calculated according to the Equation (3):TAC (mg/kg) = 615 × (A_1_ − A_2_)(3)
where 615 is the conversion factor; A1 is the absorbance of a blank sample (addition of dH_2_O); A2 is the absorbance of a sample with the addition of 15% NaHSO_3_ (*v/v*).

Total chlorophylls (TCh), chlorophyll a (Chl_a), chlorophyll b (Chl_b) and total carotenoids (TCa) were determined according to the method described by Holm [93] and Wettstein [94]. The amount of 0.2 g ± 0.01 fresh plant material was weighed, and an additional total volume of 15 mL acetone (p.a.) was added in three portions; after each, the samples were homogenized using a laboratory homogenizer (IKA, UltraTurax T-18, Staufencity, Germany). The final solutions were filtered and absorbance values were measured spectrophotometrically (Shimadzu, 1900i, Kyoto, Japan) at wavelengths of 662, 644 and 440 nm using acetone (p.a.) as a blank. The obtained absorbance values were included in the Holm–Wettstein Equations (4)–(7):Chl_a = 9.784 × A_662_ − 0.990 × A_644_ (mg/L)(4)
Chl_b = 21.426 × A_644_ − 4.65 × A_662_ (mg/L)(5)
TCh = 5.134 × A_662_ + 20.436 × A_644_ (mg/L)(6)
TCa = 4.695 × A_440_ − 0.268 × TCh (mg/L)(7)

Final results for the pigment compounds content were expressed in mg/g.

### 4.4. Determination of Antioxidant Capacity

Antioxidant capacity was determined by the ABTS assay, described by Miller et al. [95], using the 2,2′-azinobis (3-ethylbenzothiazoline-6-sulfonic acid) radical cation. This method was used because it is fast and simple to perform and is one of the most widely used for estimating antioxidant capacity. In the presence of the antioxidant, the ABTS^·+^ radical cation is reduced to ABTS and the reaction is manifested by discoloration of the blue–green solution. To prepare a stable ABTS^·+^ solution, 88 µL of a 140 mM K_2_S_2_O_8_ solution was mixed with 5 mL of ABTS solution and left in the dark for 12–16 h at room temperature. For the analysis, a 1% ABTS solution in 96% ethanol was prepared and absorbance was measured at 734 nm. Directly in a cuvette, 160 µL of the extract was mixed with 2 mL of a 1% ABTS^·+^ solution and after 5 min the absorbance was measured spectrophotometrically (Shimadzu, 1900i, Kyoto, Japan) at 734 nm. As a blank, 96% ethanol was used. Trolox (6-hydroxy-2,5,7,8-tetramethylchroman-2-carboxylic acid, Sigma-Aldrich, St. Louis, MO, USA) was used as the antioxidant standard. Final results were calculated based on a calibration curve and expressed as µmol TE/L.

### 4.5. Statistical Analyses

The obtained results were statistically analyzed using SAS software system, version 9.3. [96]. All laboratory analyses were performed in triplicate. Results were subjected to one-way analysis of variance (ANOVA) and expressed as means. Mean values were compared by t-test (LSD) and considered significantly different at *p* ≤ 0.0001. In addition to the results, different letters are given in the tables to indicate significant statistical differences between the different flower species at *p* ≤ 0.0001. The average deviations of the results from the mean for each studied parameter with the values of standard deviation (SD) are also indicated.

## 5. Conclusions

The utilization of edible flowers in the daily diet can encourage creativity, add a new appearance to culinary creations and increase the value of dishes. In addition to the fact that flowers aesthetically enhance food, their beneficial effects on human health are even more important. In this study, the content of specialized metabolites and antioxidant capacities of selected organically cultivated edible flower species were analyzed and results showed the presence and high values of phytochemicals for some flowers, although these species were not primarily considered nutritionally important. The highest AsA, as well as the highest anthocyanin content, was recorded for the cornflower. Additionally, the highest content of polyphenolic compounds and carotenoids was determined for African marigold, while the common marigold flower was the most abundant in chlorophylls. The antioxidant capacity was the lowest for zucchini flowers, while it was the same for the rest of the studied edible flower species. Hopefully this research will contribute to the popularization of this type of nutrient-rich, healthy and varied food. However, some essential information such as proper taxonomy and toxicological profiles, as well as the development of cultivation and processing methods, are still necessary for the commercialization of edible flowers.

## Figures and Tables

**Figure 1 plants-11-02529-f001:**
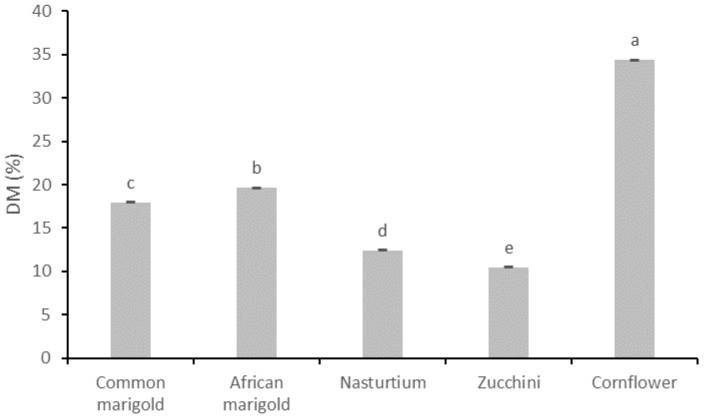
Total dry matter content (DM %) of fresh edible flowers. Results are expressed as mean ± standard deviation. Different letters indicate significant differences between mean values.

**Figure 2 plants-11-02529-f002:**
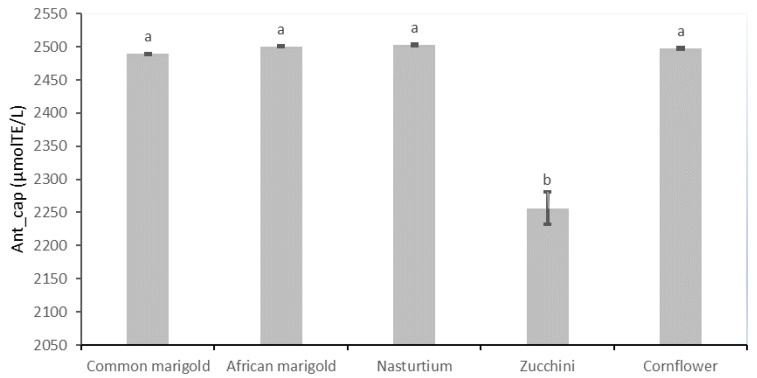
Antioxidant capacity (µmol TE/L) of fresh edible flower species. Results are expressed as mean ± standard deviation. Different letters indicate significant differences between mean values.

**Table 1 plants-11-02529-t001:** Chromaticity parameters of fresh edible flowers.

Sample	L*	a*	b*	C*	h^o^
Common marigold	70.90 ^a^ ± 4.71	26.48 ^a^ ± 6.27	73.14 ^a^ ± 8.17	80.12 ^a^ ± 3.70	72.89 ^b^ ± 5.79
African marigold	38.40 ^c^ ± 8.40	25.54 ^a^ ± 9.80	37.79 ^c^ ± 8.87	53.90 ^b^ ± 12.81	67.17 ^b^ ± 9.00
Nasturtium	53.80 ^b^ ± 8.59	28.40 ^a^ ± 9.32	49.39 ^b^ ± 2.42	51.11 ^bc^ ± 9.95	48.60 ^c^ ± 1.75
Zucchini	69.49 ^a^ ± 4.78	10.84 ^b^ ± 2.24	38.16 ^c^ ± 4.29	39.69 ^c^ ± 4.74	79.34 ^b^ ± 8.82
Cornflower	36.12 ^c^ ± 3.92	12.54 ^b^ ± 0.46	−2.49 ^d^ ± 0.83	11.12 ^d^ ± 2.28	338.35 ^a^ ± 15.80
ANOVA	*p* ≤ 0.0001	*p* ≤ 0.0001	*p* ≤ 0.0001	*p* ≤ 0.0001	*p* ≤ 0.0001
LSD	11.641	12.265	10.616	14.192	17.165

L*—lightness; a*—green–red components; b*—blue–yellow components; C*—chroma, h°—hue. Results are expressed as mean ± standard deviation. Different letters indicate significant differences between mean values.

**Table 2 plants-11-02529-t002:** Specialized metabolites content of fresh edible flowers.

Sample	AsA (mg/100 g fw)	TPC (mg GAE/100 g fw)	TNFC (mg GAE/100 g fw)	TFC (mg CTH/100 g fw)
Common marigold	25.46 ^c^ ± 0.94	379.38 ^c^ ± 0.25	213.89 ^c^ ± 1.68	165.49 ^d^ ± 1.87
African marigold	36.69 ^c^ ± 1.96	898.19 ^a^ ± 6.93	507.11 ^a^ ± 1.56	391.09 ^a^ ± 5.40
Nasturtium	77.56 ^b^ ± 0.17	336.96 ^d^ ± 0.44	138.27 ^d^ ± 0.61	198.70 ^c^ ± 1.05
Zucchini	28.69 ^c^ ± 3.61	110.24 ^e^ ± 0.82	36.63 ^e^ ± 0.69	73.59 ^e^ ± 0.51
Cornflower	129.70 ^a^ ± 11.77	647.09 ^b^ ± 0.41	382.20 ^b^ ± 1.18	264.90 ^b^ ± 1.22
ANOVA	*p* ≤ 0.0001	*p* ≤ 0.0001	*p* ≤ 0.0001	*p* ≤ 0.0001
LSD	14.466	8.1079	3.1674	6.8978

AsA—ascorbic acid; TPC—total phenolic content; TNFC—total non-flavonoid content; TFC—total flavonoid content. Results are expressed as mean ± standard deviation. Different letters show significant statistical differences between mean values with *p* ≤ 0.0001.

**Table 3 plants-11-02529-t003:** Pigment compounds of fresh edible flowers.

Sample	Chl_a (mg/g)	Chl_b (mg/g)	TCh (mg/g)	TCa (mg/g)	TAC (mg/kg)
Common marigold	0.32 ^a^ ± 0.02	0.43 ^a^ ± 0.04	0.75 ^a^ ± 0.06	0.42 ^b^ ± 0.01	nd
African marigold	0.12 ^b^	0.20 ^b^ ± 0.01	0.32 ^b^ ± 0.01	0.58 ^a^ ± 0.01	nd
Nasturtium	0.03 ^c^	0.03 ^c^	0.06 ^c^ ± 0.01	0.28 ^c^ ± 0.01	nd
Zucchini	0.03 ^c^	0.03 ^c^ ± 0.01	0.04 ^c^ ± 0.01	0.28 ^c^ ± 0.01	nd
Cornflower	nd	nd	nd	nd	1012.09 ^a^ ± 3.55
ANOVA	*p* ≤ 0.0001	*p* ≤ 0.0001	*p* ≤ 0.0001	*p* ≤ 0.0001	*p* ≤ 0.0001
LSD	0.0189	0.0438	0.0675	0.0094	4.1091

Chl_a—chlorophyll a; Chl_b—chlorophyll b; TCh—total chlorophylls; TCa—total carotenoids; TAC—total anthocyanin content; nd—not determined. Results are expressed as mean ± standard deviation. Different letters show significant statistical differences between mean values with *p* ≤ 0.0001.

## Data Availability

The data presented in this study are available in article and Appendix A.

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
