# Peer review of "Edible Flower Species as a Promising Source of Specialized Metabolites"

_plants, 2022, doi:10.3390/plants11192529_

Round 1
Reviewer 1 Report
The authors studied the flowers of 5 plant species for their content of 'specialized metabolites' and antioxidant activity to imply that those edible flowers have favorable impact on human health when consumed. Here are some comments for the purpose of improving the manuscript:
1) I think the authors mean secondary metabolites, which indeed are specific to plants, when they use the phrase 'specialized metabolites'. It would be better to define their term and mentioned a few details about plant secondary metabolites. On line 45, the authors use 'specialized metabolites and antioxidants'. What is the difference? Antioxidants are secondary metabolites and therefore specialized metabolites.
2) The Introduction section is a bit long and deviates from the objective of the research in some places. I suggest taking out text from line 92 (from 'while soil, water, ...) to 112 since is not relevant.
3) On line 130, 'highest content' for a* values reported for Nasturtium is not quite correct since the Table 1 shows that there is no significant difference in the a* values among the first three flowers. Please, correct.
4) On lines 284-285, the authors claimed that ascorbic acid are involved in the biosynthesis of ethylene, gibberellin acid and abscisic acid citing reference 60. They should make clear the involvement of ascorbic acid in the biosynthesis of those plant hormones, if any, so readers do not get the impression that ascorbic acid is a precursor of the hormones.
5) Since Socha et al. investigated the same flowers that are the object of this manuscript, why did the authors decide to study the same flowers? How is their study different than Socha's? What is the novelty brought by this manuscript?
6) Regarding the antioxidant activity of the flowers, the authors should explain why they use only on type of assay when there are so many that other laboratories publish, and why specifically ABTS assay.
7) For zucchini, Cucurbita pepo is not enough. The authors need to provide the variety of C. pepo that represents zucchini.
8) Other observations, mainly regarding the text in general and English:
a) Text just below the graphs for figures should be part of the figure legend. Place it after the figure title.
b) Make sure the names of plant species and their families are Italicized.
c) There are some sentences that need attention, since they do not make much sense. One is on lines 235-237 starting with Demasi et al. The last part of that sentence does not make English sense. Another one is on lines365-366, "These are annual plants whose zucchini seedling planting and ...".
d) "didn't" should be 'did not' and 'chemical analyzes' should be corrected.
Author Response
Dear Reviewer, In the attached document we have indicated the changes we have made based on your suggestions.

Reviewer 2 Report
It is attached.

Author Response
Dear respectable Reviewer,
thank you very much for your valuable comments and suggestions. We used the "Track Changes" option and by yellow colour we highlighted the parts of the main text we have corrected according to your valuable suggestions.
Reviewer 3 Report
The manuscript by Dujmović et al., describes the properties of eating flowers. The manuscript is relatively well written, the methodology is described in detail, and the results are sufficiently described.
However, I would like to ask the authors for a better justification of why these species were studied, what is the novelty of the presented research, as well as a better justification for undertaking research on the content of chlorophyll in flowers, and the discussion of these results.
Author Response
Dear reviewer, thank you very much for your valuable comments. We have done our best to provide answers, in addition to clarifying things that were not clear enough and taking your suggestions into consideration. Please take a look at the attached document.
